# The Effects of a Therapeutic Strategy Guided by Lung Ultrasound on 6-Month Outcomes in Patients with Heart Failure: Results from the EPICC Randomized Controlled Trial

**DOI:** 10.3390/jcm11164930

**Published:** 2022-08-22

**Authors:** Juan Torres-Macho, Jose Manuel Cerqueiro-González, Jose Carlos Arévalo-Lorido, Pau Llácer-Iborra, Jose María Cepeda-Rodrigo, Pilar Cubo-Romano, Jose Manuel Casas-Rojo, Raúl Ruiz-Ortega, Luis Manzano-Espinosa, Noel Lorenzo-Villalba, Manuel Méndez-Bailón

**Affiliations:** 1Internal Medicine Department, Hospital Universitario Infanta Leonor-Virgen de la Torre, 28031 Madrid, Spain; 2Internal Medicine Department, Hospital Universitario Lucus Augusti, 27003 Lugo, Spain; 3Internal Medicine Department, Hospital Universitario de Badajoz, 06085 Badajoz, Spain; 4Internal Medicine Department, Hospital Universitario Ramón y Cajal, 28034 Madrid, Spain; 5Internal Medicine Department, Hospital Vega Baja, 03314 Orihuela, Spain; 6Internal Medicine Department, Hospital Universitario Infanta Cristina, 28981 Madrid, Spain; 7Service de Médecine Interne, Diabète et Maladies Métaboliques, Hôpitaux Universitaires de Strasbourg, 67000 Strasbourg, France; 8Internal Medicine Department, Hospital Universitario Clínico San Carlos, 28040 Madrid, Spain

**Keywords:** heart failure, lung ultrasound, diuretic

## Abstract

Background: Pulmonary congestion (PC) is associated with an increased risk of hospitalization and death in patients with heart failure (HF). Lung ultrasound is highly sensitive for detecting PC. The aim of this study is to evaluate whether lung ultrasound-guided therapy improves 6-month outcomes in patients with HF. Methods: A randomized, multicenter, single-blind clinical trial in patients discharged after hospitalization for decompensated HF. Participants were assigned 1:1 to receive treatment guided according to the presence of lung ultrasound signs of congestion (semi-quantitative evaluation of B lines and the presence of pleural effusion) versus standard of care (SOC). The primary endpoint was the combination of cardiovascular death, readmission, or emergency department or day hospital visit due to worsening HF at 6 months. In September 2020, after an interim analysis, patient recruitment was stopped. Results: A total of 79 patients were randomized (mean age 81.2 +/− 9 years) and 41 patients (51.8%) showed a left ventricular ejection fraction >50%. The primary endpoint occurred in 11 patients (29.7%) in the SOC group and in 11 patients (26.1%) in the LUS group (log-rank = 0.83). Regarding nonserious adverse events, no significant differences were found. Conclusions: LUS-guided diuretic therapy after hospital discharge due to ADHF did not show any benefit in survival or a need for intravenous diuretics compared with SOC.

## 1. Introduction

Pulmonary congestion is the most important cause of hospital admission in patients with heart failure (HF), and it is a primary target during acute therapy. However, the clinical assessment of pulmonary congestion is often limited by the low sensitivity and specificity of physical examination and chest X-ray [1]. The greatest mortality and rehospitalization rates occur during the first weeks after a hospitalization due to HF [1,2,3]. Moreover, there is a high proportion of patients with residual pulmonary congestion at discharge [4,5], who have a higher risk of death or early readmission [6].

Pleural effusion and B-line detection with lung ultrasound (LUS) have been shown to be valid semiquantitative methods for evaluating congestion in patients with HF [7,8,9]. This information is useful for predicting short- and mid-term prognosis and is also sensitive for assessing intravascular volume variation [10,11,12,13,14,15,16,17,18,19].

According to these results, LUS-guided therapy could be useful for improving the prognosis in patients with HF. There are previous studies that suggest promising results in selected populations [20,21,22,23]. However, it has not been analyzed if its effectiveness varies in older patients. The aim of this study is to evaluate if LUS-guided diuretic therapy could improve short- and mid-term prognosis compared with conventional treatment after discharge from an acute HF hospitalization.

## 2. Materials and Methods

The EPICC (Ecografía pulmonar en la insuficiencia cardiaca crónica agudizada) study is a randomized, multicenter, single-blind clinical trial in patients with chronic HF discharged after an episode of HF decompensation. Patients were blinded for the LUS examination with the ultrasound machine turned off.

The study protocol was previously published [24]. The inclusion criteria were: age older than 18 years, NYHA functional class ≥ II at inclusion and patient’s possibility to attend ambulatory follow-up visits. The exclusion criteria were life expectancy < 6 months due to a different medical condition from HF, heart transplantation, acute coronary syndrome, recent coronary revascularization, valve replacement or resynchronization in the prior 3 months, pregnancy, restrictive pulmonary disease or severe COPD needing continuous oxygen, serum creatinine > 3 mg/dL or chronic renal insufficiency in dialysis, severe valve stenosis, ventricular arrhythmias, ICD (implantable cardioverter-defibrillator), or participation in another randomized study.

HF was defined according to the 2016 ESC guidelines for the diagnosis and treatment of acute and chronic heart failure [25]. Patients were selected within 36 h before hospital discharge. Participants were assigned 1:1 to receive medical treatment guided by the presence of congestion evaluated by LUS or conventional clinical assessment.

LUS evaluation was performed according to a previously described protocol [11,16,20]. A low-frequency convex probe (3.5–5 MHz) with abdominal configuration and 10–15 cm depth was used. During LUS examination, the screen was not visible to the patient. Each investigator had performed at least 20 previous LUS examinations.

In patients assigned to LUS-guided treatment, a diuretic dose was titrated according to a previously established protocol for evaluating the bilateral presence of B-lines in one pulmonary region and/or significant pleural effusion (>1 cm). In the standard of care (SOC) management group, the diuretic dose was adjusted according to signs and symptoms of clinical congestion and chest X-ray, as in usual clinical practice.

Both groups were treated according to the ESC Heart Failure guidelines [20]. Follow-up visits were scheduled at 7–14 days and 1, 3 and 6 months after discharge. Optional appointments were scheduled according to the patient’s clinical condition and the clinician’s choice, especially if the diuretic dose was adjusted.

The primary endpoint was the combination of cardiovascular death, readmission to hospital, emergency department visit due to HF or the need for intravenous diuretic administration at day hospital due to worsening HF at 6 months. As secondary objectives, we evaluated the differences in quality of life between the two arms through the Kansas City questionnaire assessment, and we also evaluated the sensitivity and specificity of NT-proBNP through ROC curve analysis with respect to the primary endpoint.

Cardiovascular death was defined as a composite of death due to HF, acute myocardial infarction, pulmonary embolism, stroke, sudden cardiac death or life-threatening arrhythmia. Readmission due to HF was defined as an acute hospitalization requiring more than 24 h and caused by a substantial worsening of the signs and/or symptoms of HF requiring the administration of intravenous diuretics or vasodilators.

Demographic variables, past medical history, etiology of heart failure, left ventricular ejection fraction, electrocardiogram, medical comorbidities and physical examination were recorded. Pulmonary and systemic congestion was recorded as described in the EVEREST study [25]. Analytical variables such as hemogram, sodium, potassium, serum creatinine and glomerular filtration, liver function tests and natriuretic peptides were registered.

Pulmonary congestion on LUS examination was defined as the presence of at least one positive region bilaterally and/or the presence of pleural effusion > 1 cm [22].

Drugs doses, modifications and side effects during follow-up were recorded. Side effects were defined as follows: symptomatic hypotension (a systolic blood pressure < 90 mmHg associated with symptoms justifying medical treatment adjustment) and worsening renal failure requiring an adjustment of treatment.

A sample size of 152 patients was estimated (76 patients in each arm). Based on previous Spanish studies [26], we considered an incidence of 43% for the combined variable in the SOC group versus 20% in the LUS-guided treatment group with a statistical power of 90% (ß = 0.10), an expected drop-out rate of 10% and a level of significance ∞ = 0.05 bilaterally.

Quantitative variables were expressed as means and standard deviation or median and interquartile range if they did not comply with the principles of normality. The comparison of both groups was performed using the Student *t*-test or the Mann–Whitney U test depending on the distribution of the variable. Discrete variables were compared using the chi-square test or Fisher’s exact test. We performed a ROC curve analysis with the area under the curve for NT-proBNP values and the presence of readmission and cardiovascular death. We also determined the cut-off point for NT-proBNP with the highest sensitivity and specificity.

All statistical comparisons were made according to the intention-to-treat principle. The time to the first event of the composite endpoint was considered. The Kaplan–Meier method and the log-rank test were used to compare the differences between groups. Univariable risk ratios were estimated with a Cox proportional hazards regression test. A bilateral *p*-value < 0.05 was considered statistically significant. The statistical analysis was carried out using the SPSS 17.0 software (IBM Corp., Armonk, NY, USA).

Due to COVID-19 surges, the study protocol included an interim analysis when 50% of the target population had been included. The trial could be stopped for: (1) superiority; (2) futility with regard to the primary endpoint or (3) safety reasons. Following the results of the interim analysis presented in this article, the Scientific Committee decided to prematurely stop the clinical trial, based on the futility analysis and the drop in recruitment, in February 2021.

### Ethical Aspects

This study was conducted according to the standards of the Declaration of Helsinki of 2002. All patients signed a written informed consent document. The study was approved by the Ethical Committee of Puerta de Hierro University Hospital in Madrid (2018/28981) and it was registered at ISRCTN with the number 95788878.

In September 2020, after the first and second COVID-19 surges and due to the difficulties of patient follow-up, the steering committee decided to perform an interim analysis, finishing patient recruitment.

## 3. Results

From September 2018 to September 2020, 86 patients were assessed for eligibility. Finally, 79 patients were randomized (37 in the SOC group and 42 in the LUS group). The patient selection flowchart is shown in Figure 1.

Baseline clinical and analytical characteristics and medical treatments are shown in Table 1. Both groups were balanced concerning baseline clinical and analytical characteristics. Mean age was 82.8 +/− 6.9 years in SOC group and 79.8 +/− 10.2 years in the LUS group (*p* = 0.131), and 45.5% of patients were male. Mean Everest score was 2.15 points (2.22 vs. 2.11 in the SOC and LUS groups, respectively; *p* = 0.8). Mean body mass index (BMI) in the control group and the ultrasound group was 28.5 +/− 5 vs. 28.4 +/− 4.7 respectively; *p* = 0.96. The LUS examination results at patient’s discharge are shown in Table 1. Finally, 18 patients (46.1%) were considered positive for ultrasound lung congestion. In the ultrasound group, 24 patients were positive for congestion (at least 1 positive zone bilaterally). Among them, 5 patients also showed pleural effusion 82.8%).

Patients with positive LUS examination for congestion showed no differences in the primary endpoint at six months compared with patients without them (27.7 vs. 25%; *p* = 0.8).

Table 2 shows the mean furosemide doses administered.

Regarding the relationships between natriuretic peptides and LUS, we did not find any statistically significant association between patients with a positive LUS for congestion and basal median NT-proBNP levels compared with patients with LUS without signs of congestion (4352.5 vs. 6019.5; *p* = 0.14). Using ROC curves, basal NT proBNP levels showed an AUC of 0.8 for detection of congestion in LUS. Using a cut-off value of 3271 pg/mL, we found a sensitivity of 80% and specificity of 60% (Appendix A).

NT-proBNP showed a 6-month median decrease of 1.210 pg/mL in the control group and 3.702 pg/mL in the LUS group (*p* = 0.26). The time course of the NT-proBNP levels and the B-lines during the study follow-up are shown in Appendix A.

In the LUS group, patients with signs of congestion on ultrasound examination showed a 6-month median decrease in NT-proBNP levels of 4.401 pg/mL compared with patients with no signs of congestion, who showed a slightly above median increase (355 pg/mL); *p* = 0.038.

### 3.1. Study Outcomes

The primary composite endpoint occurred in 11 patients (29.7%) in the SOC group and in 11 patients (26.1%) in the LUS group (log-rank = 0.83). At 180 days, there were no clinical or statistically significant differences in the combined endpoint, and we only found statistically significant differences in the need for intravenous furosemide ambulatory administration due to ADHF at day hospital (See Table 3). Survival analysis showed no significant differences (Figure 2).

No differences between groups (LUS and SOC) were observed on the Kansas City questionnaire at the end of the study (Appendix A). However, statistically significant differences were found in the LUS group at 6 months between patients with ultrasound signs of congestion and patients without them (70.4 +/− 17.6 vs. 45.9 + 7−15.4); *p* = 0.017).

### 3.2. Safety

Regarding nonserious adverse events, 9 events occurred in the SOC group and 14 in the LUS group. No significant differences were found among groups (Table 4).

Futility analysis showed no evidence of significant differences between groups (Z = −0.35, *p* = 0.72); boundary values: α = −2.76; β = −0.42 (Appendix A).

## 4. Discussion

The routine incorporation of LUS during HF patient follow-up may allow the for detection of subclinical pulmonary congestion, a condition associated with an increased risk of adverse effects [10,11]. Moreover, LUS is a non-radiating technique that can be performed as many times as necessary during a patient’s follow-up. In this pilot randomized controlled trial in patients discharged from hospital after an episode of acute decompensated HF, we did not obtain significant differences in relation to the primary combined endpoint between guided diuretic therapy through LUS and SOC management.

In this sense, we must point out that all patients were included in a multidisciplinary heart failure care program in Spain (UMIPIC). This clinical program has already been shown to have a significant impact on the prevention of admissions and mortality in the profile of patients included in the EPICC study. This finding seems relevant to us in the sense that many decompensations are treated as outpatients, thus preventing the patient from going to the emergency room and ending up being admitted to hospital. However, we found a greater proportion of patients with left ventricular systolic dysfunction compared with the UMIPIC population [26].

It has been described that B-line measurement is associated with a relative improvement in risk assessment at discharge following HF hospitalization when it is added to other significant prognostic variables like NYHA class or BNP [12]. The role of natriuretic peptide measurement compared with LUS-guided therapy in the management of HF patients has not been directly compared. However, the three main RCTs that have evaluated the role of LUS in guiding HF therapy measured NT-proBNP levels and found conflicting results. Rivas-Lasarte et al. and Araiza-Garaygordobil et al. found that there were no statistically significant differences at six months in NTproBNP decrease between SOC and LUS groups [21,23]. These similarities were observed despite a higher rate of loop diuretic prescription in the LUS group, whereas Marini et al. found a statistically significant reduction in NT-proBNP at 90-day follow-up in the LUS group [22]. These results could be related to the limited statistical power of the studies.

In relation to the prognostic value of NT-proBNP in this study, through ROC curve analysis, the probability of predicting admission and cardiovascular death this was evaluatedm and we observed that values above 3000 pg/mL were quite likely to predict the primary endpoint. These results are similar to previous reports [27,28,29].

Our findings show that the use of LUS-guided therapy is safe in a group of patients with significant comorbidities.

Three previous randomized clinical trials (RCT) investigating LUS-guided management of HF patients in outpatient settings that included a total of 493 patients were analyzed in a recent meta-analysis showing that this strategy significantly improved urgent visits for worsening HF at three months in one study and six months in the other two; however, it did not significantly decrease hospitalization (RR 0.65; 95% CI 0.34–1.22; *p* = 0.18) or all-cause mortality rates (RR 1.39; 95% CI 0.68–2.82; *p* = 0.37). These studies were single-center trials performed in cardiology departments, and patients were significantly younger, with fewer comorbidities and a greater proportion of left ventricular dysfunction than ours [30]. Moreover, the proportion of patients who reached the primary endpoint in our study was similar to the LUS group in a previous RCT and lower than patients in the SOC groups. These clinical aspects may have influenced our findings. Similar to our study, previous reports did not show safety concerns regarding the risks of hypokalemia or kidney injury.

Our study has several limitations. We must recognize that the trial did not reach the estimated sample size. The COVID-19 pandemic made it difficult to monitor and include patients throughout 2020 and 2021, so an interim analysis was performed, and its results led the Scientific Committee to stop recruitment. This fact limited the number of events observed and partially justifies why we found no differences between the two groups. On the other hand, lack of blinding may also have introduced bias into either treatment arm. We were unable to perform LUS in the SOC group because the attending physician and the ultrasound operator was the same person in all research centers, and therefore, we do not know the degree of congestion in this group of patients or how it could have influenced the interpretation of our results. It is possible that the long-term follow-up design of our study could have made it difficult to detect the real impact of LUS-guided therapy. However, a recent randomized clinical trial including patients admitted to the emergency department due to ADHF, determined that LUS-guided therapy conferred no benefit compared with usual care in reducing the number of B-lines at 6 h or in 30 days alive or out of hospital [31]. Up to a third of patients with heart failure have congestion at hospital discharge, which can influence readmission and mortality in these patients in the short and medium terms. In this sense, we do not know if the patients whose diuretic treatment was guided only by clinical parameters had less congestion at discharge, which could have influenced the results of our research. It seems interesting to include this line of research during depletive treatment in patients with acute heart failure from the patient’s arrival at the emergency department until discharge and to assess whether lung ultrasound is better than the standard of care.

The most important findings in our research lie in enrolling in a clinical trial of pulmonary ultrasound real-world heart failure patients admitted to internal medicine. Despite the limitations in relation to the sample size of our study, we believe that the findings are relevant from the clinical point of view in relation to the therapeutic implementation of pulmonary ultrasound in guided diuretic treatment in this patient profile. These results are consistent with the lesser importance observed in multicenter registries carried out in our internal medicine departments in our country, where it is once again demonstrated that the parameters of pulmonary congestion detected by the presence of B lines have lower prognostic sensitivity and specificity than the evaluation of the inferior vein cava [32]. In this sense, for future investigation, early effort to avoid fluid retention and timely medical treatment adjustment therapy can prevent heart failure-related hospitalizations. In this sense, in addition to the use of LUS, pro-BNP and clinical evaluation, a multisensory cardiac implantable electronic device (CIED)-based algorithm, HeartLogicTM, was created to alert in case of impending fluid retention. Felten et al. demonstrated through this method that higher and persistent alerts are indicative for true positive alerts and higher index values are indicative for more severe fluid retention [33].

## 5. Conclusions

LUS-guided diuretic therapy in patients with a recent hospital admission due to ADHF did not show any improvements in survival, readmission to hospital, ED visits or the need for intravenous diuretic over SOC.

## Figures and Tables

**Figure 1 jcm-11-04930-f001:**
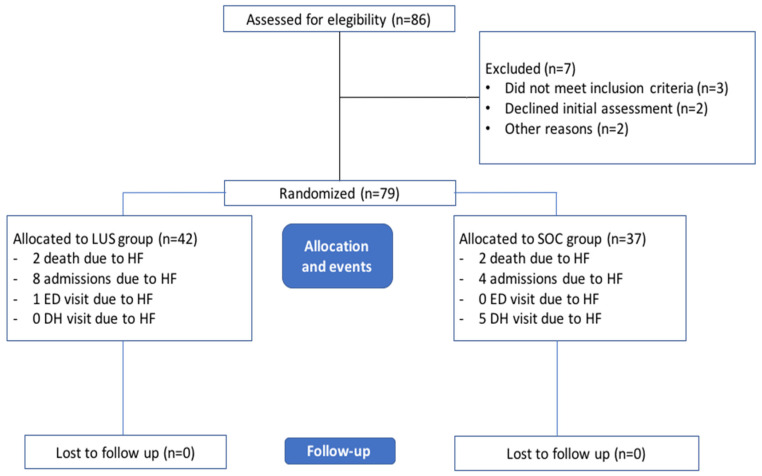
Patient selection flowchart. LUS, lung ultrasound; SOC, standard of care management group.

**Figure 2 jcm-11-04930-f002:**
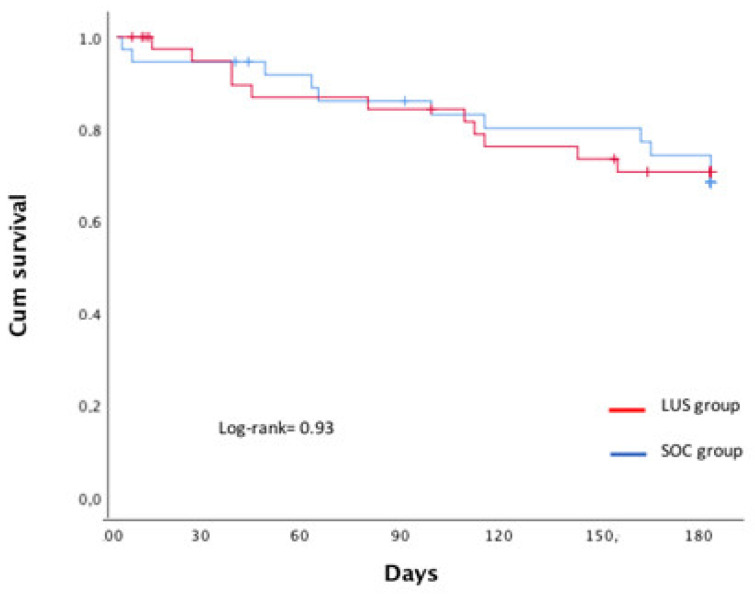
Survival analysis between the two arms at the EPICC trial.

**Table 1 jcm-11-04930-t001:** Baseline clinical characteristics, medical treatments and LUS examination results.

Variable	Overall Sample (*n* = 79)	SOC Group (*n* = 37)	LUS Group (*n* = 42)	*p* Value
Age (years) (mean, SD)	81.2 (8.9)	82.8 (6.9)	79.8 (10.2)	0.13
Male (*n*,%)	36 (45.5)	17 (45.9)	19 (45.2)	0.7
Barthel index (*n*,%)	81.8 (20.4)	82.6 (21.3)	81.1 (19.9)	0.7
HF aetiology				
Ischemic (*n*,%)	20 (25.3)	9 (24.3)	11 (26.1)	0.2
Hypertension (*n*,%)	38 (52.7)	18 (48.6)	20 (47.6)	0.1
Dilated (*n*,%)	4 (5.5)	1 (2.7)	3 (7.1)	0.09
Alcohol (*n*,%)	2 (2.5)	0 (0)	2 (4.7)	0.3
Other (*n*,%)	22 (27.8)	8 (21.6)	14 (33.3)	0.07
Comorbidities				
Hypertension (*n*,%)	69 (87.3%)	31(83.7%)	38 (90.4%)	0.2
Atrial fibrillation (*n*,%)	41(51.8%)	19(51.3%)	22(52.3%)	0.7
Diabetes Mellitus (*n*,%)	21 (26.5%)	8 (21.6%)	13 (30.9%)	0.3
CKD (*n*,%)	33 (41.7%)	13 (35.1%)	20 (47.6%)	0.2
COPD (*n*,%)	7 (8.85)	4 (10.85)	3 (7.1%)	0.5
Charlson index (*n*,%)	4.14 (2.01)	3.83 (2.04)	4.4 (2.06)	0.2
NYHA class (*n*,%)				
I	1 (1.25)	1 (2.7%)	0 (0%)	
II	35 (44.3%)	15 (40.5%)	20 (47.6%)	
III	40 (50.6%)	19 (51.3%)	21 (50%)	
IV	3 (3.7%)	2 (5.4%)	1 (2.3%)	
Everest score (*n*,%)	2.15 (1.83)	2.19 (1.85)	2.12 (1.86)	0.85
Treatment				
ACE-I/ARB (*n*,%)	77 (97.4%)	36 (97.2%)	41 (97.6%)	0.9
B-blocker (*n*,%)	64 (81%)	30 (81%)	34 (80.9%)	0.78
Loop diuretics (*n*,%)	74 (93.6%)	36 (97.2%)	38 (90.4%)	0.16
Spironolactone (*n*,%)	26 (32.9%)	12 (32.4%)	14 (33.3%)	0.11
Other diuretics (*n*,%)	13 (16.4%)	4 (10.8%)	9 (21.4%)	0.6
Digoxin (*n*,%)	3 (3.7%)	0 (0%)	3 (6.3%)	0.11
Anticoagulation (*n*,%)	44 (55.6%)	22 (59.4%)	24 (57.1%)	0.61
B-lines (>3)				
RUC			14 (35.9%)	
LUC			10 (25.7%)	
RLC			17 (53.6%)	
LLC			15 (38.4%)	
Pleural effusion (*n*, %)			8 (20.5%)	
LVEF > 50% (*n*, %)	41 (51.8)	18 (48.6%)	23 (54.7%)	0.7
Laboratory results				
NT-proBNP (pg/mL) Median (IQR 25–75)	4159 (2218–8073)	3818 (2124–7553)	4938 (2403–8780)	0.9
Creatinine (mg/dL) (mean, SD)	1.34 (0.53)	1.23 (0.51)	1.43 (0.55)	0.14
eGFR (mL/min) (mean, SD)	49.2 (20.9)	53.1 (19.8)	46 (21.5)	0.14
Haemoglobin (mg/dL) (mean, SD)	11.9 (1.7)	11.7 (1.7)	12 (1.7)	0.49

Legend: LUS: lung ultrasound; SOC: standard of care management group; CKD: chronic renal disease; COPD: chronic obstructive pulmonary disease; NYHA: New York Heart Association classification; ACE: angiotensin converting enzyme inhibitor; ARB: angiotensin-2 receptor antagonist; RUC: Right upper cuadrant; LUL: Left upper cuadrant; RLL: Right lower cuadrant; LLL: Left lower cuadrant; LVEF: left ventricular ejection fraction; IQR: interquartile range; eGFR: estimated glomerular filtration rate.

**Table 2 jcm-11-04930-t002:** Baseline and follow-up furosemide doses.

Variable	SOC Group	LUS Group	*p* Value
Diuretic Therapy titration (mg)			
MDD at baseline	76.1 (6.7)	80.3 (12.2)	0.76
MDD at visit 1 (7 days)	64.6 (5.3)	75.6 (8.3)	0.28
MDD at visit 2 (30 days)	65.6 (6.2)	80.7 (18.1)	0.4
MDD at visit 3 (90 days)	68 (6.9)	80 (18.6)	0.54
MDD at visit 4 (180 days)	74.7 (5.2)	66.6 (20.2)	0.5

Legend: LUS: lung ultrasound; SOC: standard of care management group; MDD: Mean furosemide dose.

**Table 3 jcm-11-04930-t003:** Study outcome evaluation.

Endpoint	SOC Group	LUS Group	Relative Risk	*p* Value
Admission due to ADHF	4 (10.8%)	8 (19%)	1.9 (0.53–7.06)	0.3
ED visit due to ADHF	0 (0%)	1 (2.3%)	1.96 (1.56–2.43)	0.33
Furosemide at HD	5 (13.5%)	0 (0%)	0.4 (0.33–0.56)	0.019
Death due to ADHF	2 (5.4%)	2 (4.7%)	0.87 (0.11–6.54)	0.89
Composite endpoint	11 (29.7%)	11 (26.1%)	0.83 (0.31–2.24)	0.72

Legend: LUS: lung ultrasound; SOC: standard of care management group; ADHF: acute decompensated heart failure; HD: hospital discharge.

**Table 4 jcm-11-04930-t004:** Study adverse events.

Adverse Events	SOC Group	LUS Group	*p*-Value
Acute renal failure	7 (18.9%)	9 (21.4%)	0.7
Hypotension	1 (2.7%)	4 (9.5%)	0.2
Hyponatremia	1 (2.7%)	1 (2.3%)	0.9

Legend: LUS: lung ultrasound; SOC: standard of care management group.

## Data Availability

The data presented in this study are available on request from the corresponding author. The data are not publicly available due to privacy restrictions.

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
