# Peer review of "The Effects of a Therapeutic Strategy Guided by Lung Ultrasound on 6-Month Outcomes in Patients with Heart Failure: Results from the EPICC Randomized Controlled Trial"

_jcm, 2022, doi:10.3390/jcm11164930_

Round 1
Reviewer 1 Report
I have no further comments.
Author Response
Dear reviewer,
Thank you for you comments
Reviewer 2 Report
This EPICC randomized control trial aimed to evaluate if LUS-guided diuretic therapy could improve short and mid-term prognosis compared to conventional treatment after discharge from an acute HF hospitalization, which concluded that LUS-guided diuretic therapy in patients with a recent hospital admission due to ADHF did not show any improvement in survival, readmission to hospital, ED visits or the need of intravenous diuretic compared to SOC. Probably because there were no significant differences between the 2 groups in baseline characteristics and heart failure therapies, they failed to show the superiority of LUS-guided strategy. This reviewer has some major comments as described below.
Major comments:
1. As the authors described in the Discussion section, there were 3 previous RCT of LUS-guided management in heart failure patients. This study had smaller number of enrolled patients with similar follow-up period. What was the significance of this study? What were the clinical benefits in LUS-guided heart failure management?
2. How about radiation exposure? In the non-LUS group, did the patients take chest X-ray examination instead of LUS?
Minor comment:
3. In Tables, the authors should align the digits after the decimal point. The style is not unified. Also, decimal point was mixed of “.” and “,” They should be unified.
Author Response
Dear reviewer,
Find hereby the answers to your questions
Major comments:
1. As the authors described in the Discussion section, there were 3 previous RCT of LUS-guided management in heart failure patients. This study had smaller number of enrolled patients with similar follow-up period. What was the significance of this study? What were the clinical benefits in LUS-guided heart failure management?
R/This study included older patients with an important burden of comorbid conditions compared to the 3 mentioned studies. Our mean age was was 82,8+/-6.9, females predominated and 51% of patients presented a left ventricular ejection fraction>50%. In the study conducted by Rivas et al, mean age was 69 ± 12 years and 72% were male, mean left ventricular ejection fraction was 39 ± 14%. Similarly, in the study conducted by Araiza et al, mean age was 62.5 ± 10 years, median left ventricular ejection fraction 31%
The following sentence has been added in the discussion section:
The most important findings of our research lie in enrolling in a clinical trial of pulmonary ultrasound real-world heart failure patients admitted to internal medicine. Despite the limitations in relation to the sample size of our study, we believe that they are relevant from the clinical point of view in relation to the therapeutic implementation of pulmonary ultrasound in guided diuretic treatment in this patient profile. These results are consistent with the lower importance observed in multicenter registries carried out in our internal medicine departments in our country, where it is once again demonstrated that the parameters of pulmonary congestion detected by the presence of B lines have lower prognostic sensitivity and specificity than the evaluation of the inferior vein cava.
Reference:
Pérez-Herrero, S.; Lorenzo-Villalba, N.; Urbano, E.; Sánchez-Sauce, B.; Aguilar-Rodríguez, F.; Bernabeu-Wittel, M.; Garcia-Alonso, R.; Soler-Rangel, L.; Trapiello-Valbuena, F.; Garcia-García, A.; Casas-Rojo, J.M.; Beltrán-Romero, L.; De Jorge-Huerta, L.; Molina-Puente, J.I.; Andrès, E.; Iguarán-Bermúdez, R.; Méndez-Bailón, M. Prognostic Significance of Lung and Cava Vein Ultrasound in Elderly Patients Admitted for Acute Heart Failure: PROFUND-IC Registry Analysis. J. Clin. Med. 2022, 11, 4591. https://doi.org/10.3390/jcm11154591
As stated as conclusions: LUS-guided diuretic therapy in patients with a recent hospital admission due to ADHF did not show any improve in survival, readmission to hospital, ED visits or the need of intravenous diuretic compared to SOC.
2. How about radiation exposure? In the non-LUS group, did the patients take chest X-ray examination instead of LUS?
R/ Radiation exposure was not evaluated. It is important to consider radioprotection issues. Multiple radiologic imaging exams result in an increased incidence of radiation-induced cancer in the long-term. Lung ultrasound is an important tool in diagnosis and management of different pathologies without affecting patient outcome while reducing radiation exposure.In the standard of care (SOC) management group, diuretic dose was adjusted according to signs and symptoms of clinical congestion and chest X-ray, as in usual clinical practice.
Minor comment:
3. In Tables, the authors should align the digits after the decimal point. The style is not unified. Also, decimal point was mixed of “.” and “,” They should be unified.
R/ this has been corrected
Thank you
Reviewer 3 Report
In the study by Juan Torres-Macho, they investigated “Effect of a Therapeutic Strategy Guided by Lung 2 Ultrasound on 6-Month Outcomes in Patients with Heart 3 Failure. The EPICC randomized controlled trial.” Rehospitalization after acute congestive heart failure is a poor prognostic factor, and to prevent the rehospitalization is one of the important issue. The focus of this study using the lung ultrasonic as minimum invasive method for early management after discharge is noteworthy. However, this study has some limitations and should be addressed appropriately.
Please answer the following concerns.
Concern #1
It is thought that heart failure management by a single parameter is difficult.
Boston Scientific has developed a multi-parameter method named "HeartLogic" to predict the worsening heart failure in patients with high-power device, but even this method is still controversial. Please discuss this issue.
Concern #2
The B line is a finding of pulmonary congestion as left sided heart failure, and the pleural effusion is a finding of right sided heart failure.
In this study, 20-50% of patients were discharged under congestion. Is this a reasonable condition for discharge? How was the Noria-Stevenson classification when patients discharged? Please indicate that any index of congestion at discharge does not differ between the two groups.
Also, what percentage of cases with both sided heart failure is included in this study?
Concern #3
As you already mentioned in the discussion, the population in this study is a bit unusual, with older age and more non-ischemic heart failure and CKD compared to previous studies intended to heart failure. It seems that the discussion on this point is not enough.
Why did your study become such a population? Also, please discuss that the unique population effect on outcomes in detail.
Concern #4
These are questions about patient characteristics.
There seems to be a lack of information on patient characteristics.
Please describe the clinical scenario classification at admission of two groups. This information is very important for research on AHF.
Concern #5
This study does not include body size information. The surface ultrasound image is strongly influenced by body fat, and body size can affects the outcome of heart failure therapy.
Concern #6
There was little information on arrhythmias. How many patients had ventricular arrhythmias and ICDs?
The comorbidity rate of atrial fibrillation is also important, especially in study involving many population with HFpEF.
These arrhythmias are inhibitors of heart failure therapy.
Minor comment
Table 1 is confused. Please improve it.
Author Response
Dear reviewer,
Please find hereby the answers to your questions
Concern #1
It is thought that heart failure management by a single parameter is difficult.
Boston Scientific has developed a multi-parameter method named "HeartLogic" to predict the worsening heart failure in patients with high-power device, but even this method is still controversial. Please discuss this issue.
R/ Thank you for your recommendation. The following paragraph has been added to discussion with its correspondent reference
Early detection of avoiding fluid retention and timely medical treatment adjustment therapy can prevent heart failure related hospitalizations. In this sense, besides the use of LUS, pro-BNP and clinical evaluation, a multisensory cardiac implantable electronic device (CIED) based algorithm HeartLogicTM has been created to alert in case of impending fluid retention. Felten et al, demonstrated through this method higher and persistent alerts are indicative for true positive alerts and higher index values are indicative for more severe fluid retention. We have included the reference :
- Feijen M, Egorova AD, Treskes RW, Mertens BJA, Jukema JW, Schalij MJ, Beeres SLMA. Performance of a HeartLogicTMBased Care Path in the Management of a Real-World Chronic Heart Failure Population. Front Cardiovasc Med. 2022;9:883873. doi: 10.3389/fcvm.2022.883873.
Concern #2
The B line is a finding of pulmonary congestion as left sided heart failure, and the pleural effusion is a finding of right sided heart failure.
In this study, 20-50% of patients were discharged under congestion. Is this a reasonable condition for discharge? How was the Noria-Stevenson classification when patients discharged? Please indicate that any index of congestion at discharge does not differ between the two groups.
R/ Patients were discharged according to clinical, analytical and radiological criteria by the attending physician. Ultrasound results were not taken into account in the discharge decision in the intervention group. It is remarkable that about 30% of patients who do not present clinical signs of congestion on examination, have ultrasound signs of congestion at discharge (this situation is called subclinical congestion). Eur Heart Journal (2016) 37, 1244-1251
Everest score was registered at discharge. There were no statistically significant differences between groups (2.19+/-1.83 vs 2.12 +/- 1.86; p= 0.85). This result was added to Table 1.
Also, what percentage of cases with both sided heart failure is included in this study?
R/ In the ultrasound group, 24 patients were positive for congestion (at least one positive zone bilaterally). Among them, 5 patients also showed pleural effusion 82,8%).
R/ You are right, thank you,
The following sentence has been added in the discussion section:
“up to a third of patients with heart failure have congestion at hospital discharge, which can influence readmission and mortality in these patients in the short and medium term. In this sense, we do not know if the patients who were guided in diuretic treatment only by clinical parameters had less congestion at discharge, which could have influenced the results of our research. It seems interesting to include this line of research during depletive treatment in patients with acute heart failure from the patient's arrival at the emergency department until discharge, and to assess whether lung ultrasound is better than the standard of care”.
Concern #3
As you already mentioned in the discussion, the population in this study is a bit unusual, with older age and more non-ischemic heart failure and CKD compared to previous studies intended to heart failure. It seems that the discussion on this point is not enough.
Why did your study become such a population? Also, please discuss that the unique population effect on outcomes in detail.
R/ This study included older patients with an important burden of comorbid conditions compared to the 3 mentioned studies. Our mean age was was 82,8+/-6.9, females predominated and 51% of patients presented a left ventricular ejection fraction>50%. In the study conducted by Rivas et al, mean age was 69 ± 12 years and 72% were male, mean left ventricular ejection fraction was 39 ± 14%. Similarly, in the study conducted by Araiza et al, mean age was 62.5 ± 10 years, median left ventricular ejection fraction 31%
The following sentence has been added in the discussion section:
The most important findings of our research lie in enrolling in a clinical trial of pulmonary ultrasound real-world heart failure patients admitted to internal medicine. Despite the limitations in relation to the sample size of our study, we believe that they are relevant from the clinical point of view in relation to the therapeutic implementation of pulmonary ultrasound in guided diuretic treatment in this patient profile. These results are consistent with the lower importance observed in multicenter registries carried out in our internal medicine departments in our country, where it is once again demonstrated that the parameters of pulmonary congestion detected by the presence of B lines have lower prognostic sensitivity and specificity than the evaluation of the inferior vein cava.
Reference:
Pérez-Herrero, S.; Lorenzo-Villalba, N.; Urbano, E.; Sánchez-Sauce, B.; Aguilar-Rodríguez, F.; Bernabeu-Wittel, M.; Garcia-Alonso, R.; Soler-Rangel, L.; Trapiello-Valbuena, F.; Garcia-García, A.; Casas-Rojo, J.M.; Beltrán-Romero, L.; De Jorge-Huerta, L.; Molina-Puente, J.I.; Andrès, E.; Iguarán-Bermúdez, R.; Méndez-Bailón, M. Prognostic Significance of Lung and Cava Vein Ultrasound in Elderly Patients Admitted for Acute Heart Failure: PROFUND-IC Registry Analysis. J. Clin. Med. 2022, 11, 4591. https://doi.org/10.3390/jcm11154591
As stated as conclusions: LUS-guided diuretic therapy in patients with a recent hospital admission due to ADHF did not show any improve in survival, readmission to hospital, ED visits or the need of intravenous diuretic compared to SOC.
Concern #4
These are questions about patient characteristics.
There seems to be a lack of information on patient characteristics.
Please describe the clinical scenario classification at admission of two groups. This information is very important for research on AHF.
R/ Patient´s clinical characteristics at admission were not recorded.
Concern #5
This study does not include body size information. The surface ultrasound image is strongly influenced by body fat, and body size can affects the outcome of heart failure therapy.
R/ Mean BMI in control group and ultrasound group was 28,5+/-5 vs 28,4 +/-4.7; p=0.96. This information was added
Concern #6
There was little information on arrhythmias. How many patients had ventricular arrhythmias and ICDs?
The comorbidity rate of atrial fibrillation is also important, especially in study involving many population with HFpEF.
These arrhythmias are inhibitors of heart failure therapy.
R/ Patients with ventricular arrhythmias were excluded. This information has been added
Atrial fibrillation was present in 19 patients in the control group and in 22 patients in the ultrasound group (51,3 vs 52.3%; p=0.7). This information was added in table 1
Minor comment
Table 1 is confused. Please improve it.
R/ Table has been modified
thank you
Round 2
Reviewer 2 Report
This reviewer has no further comment.
Author Response
Thank you for your comment
Reviewer 3 Report
Author Juan Torres-Macho is to be congratulated for a nicely improved article.
The author almost addressed my concerns sufficiently in this version.
However, the following a point need to be corrected.
In your response to my concern, patients with ventricular arrhythmia were excluded.
However, it is unnatural that there are no ICD patients in a study with 50% NYHA III.
Please give me a definite reason why you excluded this group from your study.
The lung ultrasound should be not disrupted by the presence of ventricular arrhythmias and implantable devices.
Author Response
Dear reviewer
Thank you for your comment
R/ ICD are less placed in Spain and these patients are followed at the Cardiologist s department and not in internal Medicine. this is the same situation for patients with ventricular arrythmias, that s why they were excluded in our study
Reference
Spanish Implantable Cardioverter-defibrillator Registry. 17th Official
Report of the Heart Rhythm Association of the Spanish Society
of Cardiology (2020). Rev Esp Cardiol. 2021;74(11):971–982
Ignacio Fernandez Lozano, Joaquın Osca Asensi, and Javier Alzueta Rodrıguez
Thank you
This manuscript is a resubmission of an earlier submission. The following is a list of the peer review reports and author responses from that submission.
Round 1
Reviewer 1 Report
In their manuscript, Torres-Macho et al. present the results of a multicentre randomized trial evaluating the prognostic effects of lung ultrasound guided diuretic treatment in patients discharged from an acute HF hospitalization. The trial did not detect any significant changes between lung ultrasound guided diuretic treatment and usual care with respect to CV death and worsening HF requiring intensified diuretic treatment. Unfortunately, the trial recruited only half of the targeted population which limits its statistical power.
The manuscript is well written and results are presented clearly, however, I have some questions:
- The authors state that the study was single blind. How was single blinding performed?
- The authors present baseline NT-proBNP measurements. Were NT-proBNP values also available at follow-up? Did NT-proBNP values guide therapy in the usual care group? Are there any studies comparing LUS-guided therapy with NT-proBNP-guided treatment?
- Table 1:
- How was CKD defined?
- Medication: there seems to be a mistake in the calculation of the percentage with beta blocker use
- mean LVEF?
- NT-proBNP values should be presented as median (IQR)
- eGFR?
- Please review the punctuation of numbers.
- Table 2: Does diuretic therapy refer to furosemide dose? Were all patients treated with furosemide, or did some receive other loop diuretics such as torasemide? Why aren't results presented for 6 months follow-up?
- Although no statistically significant, the loop diuretic dose in the LUS-group was higher at 3 months than in the usual care group. Do you have any information on change in eGFR between baseline and follow-up in both groups?
- Discussion: The discussion is quite short. I would like to know
- Is the number of patients who reached the primary endpoint lower/higher/similar to that in other comparable study cohorts?
- Are there any studies comparing LUS-guided therapy with NT-proBNP guided treatment?
- Are there any studies suggesting benefits of a LUS-guided approach in addition to natriuretic peptide measurements?
- Please provide more details on the recent meta-analysis on LUS-guided management. How many studies were included? Who many patients were included? What was the median follow-up duration? Were NT-proBNP measurements available in the included studies?
- I'm afraid I don't understand the last paragraph of the discussion section.
- Minor comment: In the first sentence of the introduction, the authors should use "target" instead of "goal".
Author Response
In their manuscript, Torres-Macho et al. present the results of a multicentre randomized trial evaluating the prognostic effects of lung ultrasound guided diuretic treatment in patients discharged from an acute HF hospitalization. The trial did not detect any significant changes between lung ultrasound guided diuretic treatment and usual care with respect to CV death and worsening HF requiring intensified diuretic treatment. Unfortunately, the trial recruited only half of the targeted population which limits its statistical power.
The manuscript is well written and results are presented clearly, however, I have some questions:
- The authors state that the study was single blind. How was single blinding performed?
Single blinding was done performing the lung ultrasound examination with the ultrasound machine turned off in the control group.
- The authors present baseline NT-proBNP measurements. Were NT-proBNP values also available at follow-up? NT-proBNP was performed systematically at admission but not during follow up and there were a lot of missing data.
- Did NT-proBNP values guide therapy in the usual care group? Guided therapy was performed using physical examination, chest-X ray and analytical parameters by the attending physicians.
- Are there any studies comparing LUS-guided therapy with NT-proBNP-guided treatment? To our knowledge there are no previous reports comparing lung ultrasound and Natriuretic peptides guided therapy.
- Table 1:
- How was CKD defined? CKD was defined as the presence of a previous glomerular filtration < 90 ml/min.
- Medication: there seems to be a mistake in the calculation of the percentage with beta blocker use. Percentage was corrected in the draft.
- mean LVEF? LVEF was registered as a dicotomic variable (defined as the presence of an EF greater or lower tan 50%.
- NT-proBNP values should be presented as median (IQR). This suggestion has been included in the table.
- eGFR? This data has been included in the table.
- Please review the punctuation of numbers. Punctuation was corrected.
- Table 2: Does diuretic therapy refer to furosemide dose? This was corrected. Were all patients treated with furosemide, or did some receive other loop diuretics such as torasemide? All patients received furosemide
- Why aren't results presented for 6 months follow-up? Information was included in the table.
- Although no statistically significant, the loop diuretic dose in the LUS-group was higher at 3 months than in the usual care group. Do you have any information on change in eGFR between baseline and follow-up in both groups? This data is not avaliable.
Discussion: The discussion is quite short. I would like to know
- Is the number of patients who reached the primary endpoint lower/higher/similar to that in other comparable study cohorts? This information was included in the discussion.
- Are there any studies comparing LUS-guided therapy with NT-proBNP guided treatment? To our knowledge there are no studies that compare LUS vs NP guided therapy.
- Are there any studies suggesting benefits of a LUS-guided approach in addition to natriuretic peptide measurements? No randomized controlled trials have analyzed this possibility.
- Please provide more details on the recent meta-analysis on LUS-guided management. How many studies were included? Who many patients were included? What was the median follow-up duration? Were NT-proBNP measurements available in the included studies? This information was included in the discussion section.
- I'm afraid I don't understand the last paragraph of the discussion section. This paragraph was deleted.
- Minor comment: In the first sentence of the introduction, the authors should use "target" instead of "goal". This minor change was included.
Reviewer 2 Report
In the EPICC randomized controlled trial Torres-Macho et al. demonstrated the lack of benefit from routinely conducted LUS in the patients discharged after HF decompensation.
The presented study is well-written. However, it has several limitations which in my opinion make it difficult to accept for publication. Firstly, due to COVID-19 pandemic, the assumed number of patients was not recruited. Most importantly, similar studies have been already published. Based on them, we already know that routine LUS screening is not recommended. LUS should be used as a useful tool during hospitalization, especially in the ICU. In my opinion, it is not enough to recruit the older population and to present adverse side effects to consider them as novel findings.
Author Response
In the EPICC randomized controlled trial Torres-Macho et al. demonstrated the lack of benefit from routinely conducted LUS in the patients discharged after HF decompensation.
The presented study is well-written. However, it has several limitations which in my opinion make it difficult to accept for publication. Firstly, due to COVID-19 pandemic, the assumed number of patients was not recruited. .
Although the study did not reach the estimated sample size, an interim futility analysis showed that there were no evidences of significant differences between groups
Most importantly, similar studies have been already published. Based on them, we already know that routine LUS screening is not recommended. LUS should be used as a useful tool during hospitalization, especially in the ICU. In my opinion, it is not enough to recruit the older population and to present adverse side effects to consider them as novel findings
A recent metanalysis that is mentioned in the discussion showed that LUS may reduce the number of visits to the emergency department.
Reviewer 3 Report
This is a clinical study, which aimed to evaluate if LUS-guided diuretic therapy could improve short and mid-term prognosis compared to conventional treatment after discharge from an acute HF hospitalization. The authors concluded that LUS-guided diuretic therapy in patients with a recent hospital admission due to ADHF did not show any improve in survival, readmission to hospital, ED visits or the need of intravenous diuretic compared to SOC. They failed to show the superiority of LUS-guided diuretic therapy, and this reviewer has some major comments as described below.
Major comments:
1. In the Methods section (lines 87-91), the authors described the titration of diuretics; however, more details should be written. How did the authors titrate the doses? The titration was different between LUS and SOC?
2. In Table 1, eGFR should be added.
3. In both groups, how often were chest X-ray examinations performed during the follow-up period?
4. How about the renal function worsening in the follow-up period?
5. This study did not show the benefits of LUS. However, there might be some good aspect of LUC follow up, for example, radiation exposure. They should discuss more about the good points of LUS in the Discussion section.
Minor comments:
6. In Tables, the authors should align the digits after the decimal point.
7. Also, decimal point was mixed of “.” and “,” They should unify.
8. Table 3. Relative risk of “Death due to ADHF”. “(.11-6.54)” should be “(0.11-6.54)”
Author Response
This is a clinical study, which aimed to evaluate if LUS-guided diuretic therapy could improve short and mid-term prognosis compared to conventional treatment after discharge from an acute HF hospitalization. The authors concluded that LUS-guided diuretic therapy in patients with a recent hospital admission due to ADHF did not show any improve in survival, readmission to hospital, ED visits or the need of intravenous diuretic compared to SOC. They failed to show the superiority of LUS-guided diuretic therapy, and this reviewer has some major comments as described below.
Major comments:
- In the Methods section (lines 87-91), the authors described the titration of diuretics; however, more details should be written. How did the authors titrate the doses? The titration was different between LUS and SOC? Diuretc titration was guided by the clinical impression of congestion of each physician. The only difference among groups was the information added by LUS about pulmonary congestion.
- In Table 1, eGFR should be added. This information has been included.
- In both groups, how often were chest X-ray examinations performed during the follow-up period? Chest X ray was perfomed during follow up was performed depending on physicians criteria.
- How about the renal function worsening in the follow-up period? This information is not available.
- This study did not show the benefits of LUS. However, there might be some good aspect of LUC follow up, for example, radiation exposure. They should discuss more about the good points of LUS in the Discussion section. This suggestion has been included.
Minor comments:
- In Tables, the authors should align the digits after the decimal point. Digits were aligned.
- Also, decimal point was mixed of “.” and “,” They should unify. Decimal pointing was Unified.
- Table 3. Relative risk of “Death due to ADHF”. “(.11-6.54)” should be “(0.11-6.54)”. This change has been included.
Round 2
Reviewer 1 Report
Dear authors,
Thanks for answering my questions. However, as I read the revised version of the manuscript, I noticed that unfortunately only few changes were incorporated in the manuscript. Please incorporate all aspects that are discussed above in the manuscript! Particularly, I ask you to include a paragraph that discusses the role of natriuretic peptide measurements vs. LUS in the management of HF patients. Are there any non-randomized studies suggesting a benefit of a LUS-guided apporach in addition to cardiac peptide measurements?
Also, I couldn't find any information whether the number of patients who reached the primary endpoint was lower/higher/similar to other comparable cohorts.
Please revise your manuscript accordingly. Please track changes in the revised manuscript in read colour so that revisions can be identified easily.
Author Response
Dear reviewer 1:
Thank you very much for your answer.
All the changes have now been marked in the text (highlighted in yellow and with a comment). Sorry about the inconveniences.
Related to the following request "I ask you to include a paragraph that discusses the role of natriuretic peptide measurements vs. LUS in the management of HF patients. Are there any non-randomized studies suggesting a benefit of a LUS-guided apporach in addition to cardiac peptide measurements? " the following paragraph was included in the discussion:
"It´s been described that B-lines measurement is associated with a relevant improvement of risk assessment at discharge following HF hospitalization when it is added to other significant prognostic variables like NYHA class or BNP [12]. The role of natriuretic peptide measurement compared to LUS guided therapy in the management of HF patients has not been directly compared. However, the three main RCT that have evaluated the role of LUS in guiding HF therapy measured NT-proBNP levels, finding conflicting results. Rivas-Lasarte et al and Araiza-Garaygordobil et al found that there were no statistically significant differences at six months in NTproBNP decrease between SOC and LUS group [21,23]. These similarities were observed despite a higher rate of loop diuretic prescription in the LUS group; whereas, Marini et al found a statistically significant reduction of NT-proBNP value at 90-day follow-up in the LUS group [22]. These results could be related to the limited statistical power of the studies".
If there is any other study that you consider that should be included in the discussion, please do not hesitate to let us know.
Related to the following request: "Also, I couldn't find any information whether the number of patients who reached the primary endpoint was lower/higher/similar to other comparable cohorts."
The number of patients who reached de primary endpoint was similar to the LUS group of previous RCT and lower than standard of care groups. This was included in the text.
The rest of Round 1 questions are answered with more detail than in our previous reply, explaining if the answer has been included in the text.
The authors state that the study was single blind. How was single blinding performed? Patients were blinded performing the LUS examination with the ultrasound machine turned off. This was included in the text.
The authors present baseline NT-proBNP measurements. Were NT-proBNP values also available at follow-up? NT-proBNP was performed systematically only at admission.
Did NT-proBNP values guide therapy in the usual care group? No
Table 1:
-
- How was CKD defined? CKD was defined as the presence of a previous glomerular filtration < 90 ml/min. This was included in the text.
- Medication: there seems to be a mistake in the calculation of the percentage with beta blocker use. Percentage was corrected in the Table.
- mean LVEF? Data not avaliable. The variable was registered as lower or greater than 50%.
- NT-proBNP values should be presented as median (IQR). This information was changed.
- eGFR? This data was not avaliable
- Please review the punctuation of numbers. Punctuation was corrected.
Table 2: Does diuretic therapy refer to furosemide dose? This was corrected in the table. Were all patients treated with furosemide, or did some receive other loop diuretics such as torasemide? All patients received furosemide Why aren't results presented for 6 months follow-up? Data was not avaliable at 6 months in a significant proportion of patients.
Although no statistically significant, the loop diuretic dose in the LUS-group was higher at 3 months than in the usual care group. Do you have any information on change in eGFR between baseline and follow-up in both groups? This data is not avaliable.
Discussion: The discussion is quite short. I would like to know
-
- Is the number of patients who reached the primary endpoint lower/higher/similar to that in other comparable study cohorts? Similar to the LUS group of previous RCT and lower than standard of care groups. This was included in the text.
- Are there any studies comparing LUS-guided therapy with NT-proBNP guided treatment?
- Are there any studies suggesting benefits of a LUS-guided approach in addition to natriuretic peptide measurements?
- The following paragraph was included trying to answer the previous two questions.
"It´s been described that B-lines measurement is associated with a relevant improvement of risk assessment at discharge following HF hospitalization when it is added to other significant prognostic variables like NYHA class or BNP [12]. The role of natriuretic peptide measurement compared to LUS guided therapy in the management of HF patients has not been directly compared. However, the three main RCT that have evaluated the role of LUS in guiding HF therapy measured NT-proBNP levels, finding conflicting results. Rivas-Lasarte et al and Araiza-Garaygordobil et al found that there were no statistically significant differences at six months in NTproBNP decrease between SOC and LUS group [21,23]. These similarities were observed despite a higher rate of loop diuretic prescription in the LUS group; whereas, Marini et al found a statistically significant reduction of NT-proBNP value at 90-day follow-up in the LUS group [22]. These results could be related to the limited statistical power of the studies".
Please provide more details on the recent meta-analysis on LUS-guided management. How many studies were included? Three Who many patients were included? 493. What was the median follow-up duration? 6 months in two reports and 3 months in the other. Were NT-proBNP measurements available in the included studies? Yes and this issue was included in the previous paragraph. All these answers were included in the text.
I'm afraid I don't understand the last paragraph of the discussion section. This paragraph was removed from the text.
Minor comment: In the first sentence of the introduction, the authors should use "target" instead of "goal". This minor change was added.
Reviewer 2 Report
Thank you for your response. However, as previously, I recommend the rejection of your manuscript.
Author Response
Dear reviewer 2.
Thank you very much for your answer.
Regards.
Reviewer 3 Report
This reviewer has no further comment.
Author Response
Dear reviewer 3.
Thank you very much for your answer.
Regards.